# Validation and Measurement Invariance of the Body Appreciation Scale-2 between Genders in a Malaysian Sample

**DOI:** 10.3390/ijerph182111628

**Published:** 2021-11-05

**Authors:** Chee-Seng Tan, Siew-May Cheng, Chin Wen Cong, Afi Roshezry Bin Abu Bakar, Edwin Michael, Mohamad Iqbaal Bin Mohd Wazir, Muliyati Binti Mat Alim, Bazlin Darina Binti Ahmad Tajudin, Nor Ez-Zatul Hanani Binti Mohamed Rosli, Alfian Bin Asmi

**Affiliations:** 1Department of Psychology and Counselling, Faculty of Arts and Social Science, Universiti Tunku Abdul Rahman (UTAR), Kampar 31900, Malaysia; chinwencong@1utar.my; 2Department of Languages and Linguistics, Faculty of Arts and Social Science, Universiti Tunku Abdul Rahman (UTAR), Kampar 31900, Malaysia; chengsm@utar.edu.my (S.-M.C.); iqbaal@utar.edu.my (M.I.B.M.W.); alfian@utar.edu.my (A.B.A.); 3Department of Journalism, Faculty of Arts and Social Science, Universiti Tunku Abdul Rahman (UTAR), Kampar 31900, Malaysia; afi@utar.edu.my (A.R.B.A.B.); edwinm@utar.edu.my (E.M.); 4Department of Advertising, Faculty of Arts and Social Science, Universiti Tunku Abdul Rahman (UTAR), Kampar 31900, Malaysia; muliyati@utar.edu.my; 5Department of Public Relations, Faculty of Arts and Social Science, Universiti Tunku Abdul Rahman (UTAR), Kampar 31900, Malaysia; bazlin@utar.edu.my (B.D.B.A.T.); zatulhanani@utar.edu.my (N.E.-Z.H.B.M.R.)

**Keywords:** BAS-2, body image, measurement invariance, Malaysia, selfie editing

## Abstract

The 10-item Body Appreciation Scale-2 (BAS-2) is a measurement for individuals to self-report the extent to which they accept and respect their bodies. Although the BAS-2 has been translated into the Malay language and found to have promising qualities, the psychometric characteristics of the English version of BAS-2 remain unknown in the Malaysian context. The present study thus administered the English version BAS-2 and selfie-editing frequency scale to 797 individuals aged 18 to 56 years old in Malaysia. The dataset that was randomly divided into two halves were submitted to exploratory factor analysis and confirmatory factor analysis respectively. Both of the factor analyses consistently support a one-factor model. The Cronbach’s alpha and McDonald omega coefficients were greater than 0.90, indicating that the BAS-2 has good internal consistency. The incremental validity is also evident. A hierarchical multiple regression showed that the BAS-2 score had a positive relationship with selfie-editing frequency after controlling for age and gender. Moreover, the measurement invariance test supported scalar invariance between genders, and an analysis of covariance did not find significant gender differences. Overall, the findings replicate past findings and regularly support the usability of the BAS-2 in the Malaysian context. The implications of the BAS-2 and future directions are also discussed.

## 1. Introduction

Obesity has always been a great challenge to Malaysia. According to the World Population Review [1], Malaysia recorded the highest obesity prevalence at 15.6% in Southeast Asia, followed by Brunei (14.1%) and Thailand (10.0%). Moreover, the National Health and Morbidity Survey shows that the obesity prevalence in adults keeps increasing, from 15.1% in 2011 [2], 17.7% in 2015 [3] to 19.7% in 2019 [4].

Obesity is detrimental to body satisfaction [5]. Studies have consistently shown a positive relationship between obesity and body dissatisfaction (e.g., [6,7,8,9]). Indeed, a meta-analysis concluded that obese individuals were more dissatisfied with their bodies than non-obese individuals [10]. More importantly, body dissatisfaction is associated with physical health issues such as compulsive exercise [11], sleep deprivation [12], and low psychological well-being [13]. Likewise, individuals with higher levels of body dissatisfaction are at greater risk of depressive symptoms [14], food addiction [15], and eating disorders [16]. Taken together, it is believed that the high prevalence rate of obesity is likely to increase the level of body dissatisfaction among Malaysians that would impair their physical and psychological wellness. Therefore, other than tackling obesity, it is equally important to identify factors that are helpful to instill an accurate appreciation of body image. For instance, He and colleagues [17] conducted a meta-analysis on 40 studies published between 2008 and 2019 and found a significant gender difference in body appreciation. Compared to females, males reported a higher level of body appreciation, though the effect was small (Cohen’s *d* = 0.27, 95% confidence interval (CI): 0.21, 0.33, *p* < 0.001). Moreover, the mean gender was found to negatively predict the gender differences. In other words, the gap in body appreciation between genders reduced when age increased.

To achieve the above-mentioned goal, however, a measurement tool of body appreciation is needed. Avalos and colleagues [18] developed the Body Appreciation Scale (BAS), a single-dimension self-report to measure the extent to which individuals accept and respect their bodies. However, the BAS has been found to have two dimensions in different cultural contexts. Moreover, the items also show low factor loading and carry a different meaning for women and men. Therefore, Tylka and Wood-Barcalow [19] retained the five items of the BAS and replaced the unsatisfactory items with another five new items to construct the Body Appreciation Scale-2 (BAS-2). Unlike the BAS, the BAS-2 has consistently been found to have a one-dimensional factorial structure across different nations and samples (e.g., [20], see [21] for a review). Furthermore, the BAS-2 has also been translated into different languages. The translated versions of BAS-2 show satisfactory psychometric qualities (e.g., [22]). For instance, Swami and colleagues [21] translated the BAS-2 into the Malay language and found promising psychometric properties on 761 individuals aged 18 to 63 years old in Malaysia. Specifically, the one-dimensional factorial structure was found in both the exploratory factor analysis (EFA) and the confirmatory factor analysis (CFA). The partial scalar invariance was supported across gender and ethnicity between the Malays and the Chinese participants. The Malay version of the BAS-2 score demonstrated good internal consistency (McDonald omega [ω] = 0.88) and convergent validity as well as incremental validity in predicting subjective happiness after controlling for the effects of other variables (e.g., perceived media influence).

### The Present Study

Although the Malay version of the BAS-2 shows promising psychometric qualities, little is known about the psychometric properties of the original English version of the BAS-2 in the Malaysian context. It is therefore critical to address this gap for two reasons. First, it is inadequate to generalize the findings of the Malay version of the BAS-2 in the English version of the BAS-2. Inarguably, even when a well-established measurement has been translated into many different languages, researchers are still required to examine the psychometric qualities of the translated version before employing it. This is necessary since the participants presented with items in different languages may perceive the items differently and hence the factorial structure, reliability, and validity of the measurement may be compromised [23].

Second, Malaysia is made up of four major ethnic groups, namely Bumiputera (including Malays and Indigenous peoples), Chinese, Indians, and others. Malaysians residing in their multiethnic and multilingual country freely use multiple languages in formal and informal settings [24]. Moreover, despite Malay being the official language in Malaysia, only Malays will adopt the Malay language as their mother tongue, as do the Chinese and Indians adopting Chinese and Tamil languages themselves. Furthermore, risk of bias in the item responses resulting from this difference necessitates answering the BAS-2 in the participants’ respective mother tongue in a multilingual country like Malaysia [21]. Some studies (e.g., [25,26]) have, in fact, found that items of the original English version’s measurements that have been validated elsewhere are not appropriate in the Malaysian context.

Taken together, the present study was set to examine the factorial structure (using EFA and CFA), reliability, incremental validity (using selfie-editing frequency), and tested measurement invariance of the BAS-2 using a multigroup CFA in the Malaysian context. The findings are expected to shed light on the usability of the English version of the BAS-2 in Malaysia.

## 2. Materials and Methods

### 2.1. Participants and Design

A total of 799 individuals currently residing in Malaysia participated in the present study voluntarily. Of the participants, 2 individuals below 18 years old were excluded resulting in 797 responses (60% females). The mean age was 22.97 (SD = 5.42) and it ranged from 18 to 56 years old. The sample consisted of Chinese (58%), Malays (27.20%), Indians (11.40%), and participants with other ethnicities (3.40%). The participants were recruited either from undergraduate courses at universities or social networking sites using convenience and snowball sampling. The data, which were collected between August and October 2020 for a larger project of body image, were approved by the Scientific and Ethical Review Committee of Universiti Tunku Abdul Rahman (Ref: U/SERC/104/2020). The variables that are irrelevant to the present study were not reported for the sake of clarity. After replacing the missing values (<2%) using the expectation–maximization technique (Little’s test χ^2^(1243) = 1095.527, *p* = 0.999), the dataset was randomly split into two sets: exploratory and confirmatory datasets. The exploratory dataset was submitted to an EFA (*n* = 392) to explore the factorial structure of the BAS-2, while the confirmatory dataset was used for a CFA (*n* = 405) to verify the factorial solution suggested by the EFA.

### 2.2. Measurement

The Body Appreciation Scale-2 (BAS-2; [19]) was employed. The participants were instructed to rate 10 items using a 5-point Likert-type scale with the options from 1 (never) to 5 (always). Examples of items are “I respect my body” and “I feel love for my body.” A mean score is computed by averaging all of the item scores. Higher mean scores indicate a higher level of body appreciation. The reliability of the BAS-2 was reported in the Results section.

Besides, we employed Yue et al.’s [27] 13-item index to measure the selfie-editing frequency. The relationship between body appreciation and the selfie-editing frequency was examined to test the incremental validity of the BAS-2. This is guided by the impression management theory [28] in that people have a desire for manipulating impressions on others through purposive self-exposure during social interactions. In other words, people who are unsatisfied with their body image (i.e., low in body appreciation) tend to modify their selfie images to present positive sides of themselves. Following the practice of Wang [29], we added the item “I use applications that can add filter to the camera automatically” to reflect the common practice of selfie editing nowadays. The participants indicated their agreements (1 = never to 7 = every time) on a Likert scale statement like “I slim the size of my face in my selfies.” The 14-item selfie-editing frequency score showed excellent internal consistency for the total sample in the present study: α = 0.92, 95% CI (0.91, 0.93), ω = 0.92, 95% CI (0.91, 0.93).

### 2.3. Analytic Process

The JASP software package (version 0.14.1) was used to conduct the analyses. An EFA was first conducted to investigate the potential factorial structure followed by a CFA to examine whether the model revealed by the EFA is the best fit model. Although the BAS-2 items were rated on a 5-point Likert scale, the descriptive analysis showed that about 85% of the responses fell into the categories of sometimes, often, and always. The imbalanced distribution suggests that it is more adequate to treat the responses as ordinal than continuous data. Following the suggestion of Forero et al. [30], we used the unweighted least squares (ULS) estimation method. The model fit was evaluated based on indices such as the ratio of chi-square values to degrees of freedom (χ^2^/df), comparative fit index (CFI), Tucker–Lewis Index (TLI), root mean square error of approximation (RMSEA), and the standardized root mean square residual (SRMR). A well-fitting model should pass the recommended cutoff criteria, such as χ^2^/df < 3, CFI and TLI > 0.95, RMSEA ≤ 0.05, and SRMR < 0.08 [31,32]. Besides, the internal consistency of the BAS-2 was examined using Cronbach’s alpha (α) and ω coefficients. The incremental validity was tested using the selfie-editing frequency. It is assumed that body appreciation will have a negative relationship with selfie-editing frequency.

Meanwhile, a multigroup CFA was conducted to measure the invariance between genders at the configural, metric, and scalar levels. After demonstrating the unconstrained baseline model of the two groups that shall fit the data well, the configural invariance requires the factorial structure of the BAS-2 to be similar across groups. Likewise, the metric invariance requires similarity of the magnitude of the factor loadings across the groups by comparing the model with fixed factor loading and the configural invariance model. Lastly, scalar invariance is evident if the item loadings and item intercepts are parallel across the groups. The three types of the invariance tests were evaluated using differences in chi-square (Δχ^2^) and the CFI (ΔCFI). However, as the large sample size may affect the results of Δχ^2^, we mainly refer to ΔCFI and conclude an invariance when ΔCFI is < 0.01 [33]. 

## 3. Results

### 3.1. Exploratory Factor Analysis

An EFA using a parallel analysis, principal axis factoring (PAF) estimation, and varimax rotation was run to examine the factorial structure underlying the 10-item BAS-2. As the parallel analysis suggested one factor, another EFA without employing rotation was conducted. The Kaiser–Meyer–Olkin (KMO) value was 0.94 and the Bartlett’s test of sphericity was statistically significant, χ^2^ (45) = 2336.69, *p* < 0.001. The one-factor model, χ^2^ (35) = 123.91, *p* < 0.001, explained 55.90% of the total variance, TLI = 0.950, RMSEA = 0.081, 90% CI (0.07, 0.10). All of the factor loadings were greater than 0.60, ranging from 0.66 (item 5) to 0.84 (item 6). Table 1 summarizes the results of the EFA. The BAS-2 score also showed excellent internal consistency: α = 0.93, 95% CI (0.91, 0.94), ω = 0.93, 95% CI (0.92, 0.94).

### 3.2. Confirmatory Factor Analysis

The 10-item single-factor model revealed by the EFA in Study 1 was tested using a CFA with an unweighted least squares (ULS) estimation. The model showed a good fit to the data: χ^2^ (35) = 38.19, *p* = 0.327, χ^2^/df = 1.09, CFI = 1.000, TLI = 0.999, RMSEA = 0.015, 90% CI (0.00, 0.04), and SRMR = 0.037. The standardized factor loadings were statistically significant at 0.001 level and ranging from 0.621 (item 1) to 0.828 (item 7). Similarly, the BAS-2 score exhibited excellent internal consistency: α = 0.92, 95% CI (0.91, 0.93); ω = 0.92, 95% CI [0.91, 0.93].

### 3.3. Validity 

As both the EFA and the CFA results consistently support a one-factor model, we used the total sample for testing the validity and measurement invariance between the gender groups. Prior to that, we ran a CFA on the total sample to ensure the fitness of the model to the data. The result showed a good fit: χ^2^ (35) = 57.32, *p* = 0.010, χ^2^/df = 1.64, CFI = 0.998, TLI = 0.998, RMSEA = 0.028, 90% CI = (0.01, 0.04), and SRMR = 0.032.

A hierarchical multiple regression with selfie-editing frequency as the outcome variable was conducted to examine the incremental validity of the BAS-2. As the BAS-2 score was correlated with age (*r* = 0.125, *p* < 0.001) and selfie-editing frequency (*r* = −0.087, *p* = 0.014), and there was a gender difference in age (Levene’s test = 25.66, *p* < 0.001; Welch’s *t* (511.62) = −4.14, *p* < 0.001, Cohen’s *d* = −0.31, 95% CI (−0.45, −0.17)), and selfie-editing frequency (Levene’s Test = 8.59, *p* = 0.003; Welch’s *t* (628.663) = 4.425, *p* < 0.001, Cohen’s *d* = 0.32, 95% CI (0.18, 0.47)), age and gender (female as reference) were entered to the first model (Model 1) followed by the BAS-2 score (Model 2). The Model 2 was statistically significant and explained 4.30% of the variance: *F*(3, 793) = 11.83, *p* < 0.001, Δ*R*^2^ = 0.005, Δ*F*(1, 793) = 3.93, *p* = 0.048. After controlling for the effect of age (β = −0.11, *p* = 0.003) and gender (β = −0.14, *p* < 0.001), the BAS-2 scores (β = −0.07, *p* = 0.048) showed a negative relationship with the selfie-editing frequency, supporting the incremental validity.

### 3.4. Measurement Invariance

The one-factor model showed a good fit for both male and female groups (see Table 2). Similarly, the configural invariance model, metric invariance model, and scalar invariance model demonstrated a good fit. The comparison between the CFI values of the configural invariance model and metric invariance model was within the acceptable level. The results showed support for metric invariance. Likewise, a comparison between the metric invariance model and scalar invariance model supported the scalar invariance.

Such support for scalar invariance indicates the adequacy of the testing whether there is a difference between males (*n* = 319) and females (*n* = 478) in the BAS-2 score. An analysis of covariate (ANCOVA) with age and selfie-editing frequency as the covariate variables was thus conducted to compare the BAS-2 score between the two gender groups. The results showed that both age, *F*(1, 793) = 10.28, *p* = 0.001, η^2^ = 0.01, and the selfie-editing frequency, *F*(1, 793) = 3.93, *p* = 0.048, η^2^ = 0.01, had a significant relationship with the BAS-2 score. After excluding the effects of age and the selfie-editing frequency, there was no significant difference between the males (*M* = 3.79, *SD* = 0.83) and females (*M* = 3.73, *SD* = 0.82) in the BAS-2 score, *F*(1, 793) = 0.01, *p* = 0.928.

## 4. Discussion

The current study assessed the psychometric properties of the English version of the BAS-2 (Tylka and Wood-Barcalow, 2015) among individuals in Malaysia. In line with the literature (e.g., [19]), the English version of the BAS-2 demonstrates good qualities in the Malaysian context.

We first explored the factorial structure of the BAS-2 using an EFA. A parallel analysis revealed a single factor thus supporting the BAS-2 as a unidimensional scale. Next, we conducted a CFA to confirm the unidimensionality of the (English) BAS-2. In line with the literature (e.g., [19,20,22]), the 10-item single-factor model produced good model fit indices and excellent internal consistency. The consistent results suggest that the original English version of the BAS-2 is best represented by a single factor. Notably, our results also replicate Swami and colleagues’ [21] findings of the Malay version, suggesting that translation does not impose any major threats to the psychometric characteristics of the BAS-2.

In addition, we also found evidence of the incremental validity of the BAS-2. After statistically controlling for age and gender, the BAS-2 still showed a significant and negative relationship with selfie-editing frequency. The results are in line with the impression management theory [28] that regardless of gender, individuals who accept and respect their bodies (i.e., scoring high in the BAS-2) alter their selfie images less frequently. In addition, the measurement invariance between genders is also evident. While Swami and colleagues’ [21] found a partial scalar invariance between genders in the Malay version of the BAS-2, our results support that the English version of the BAS-2 scores are (fully) scalar invariant. The findings are consistent with past studies (e.g., [20,22]) in that both language versions of the BAS-2 are equivalently applicable to males and females in Malaysia. Furthermore, the ANCOVA results showed no significant gender differences in the BAS-2 score. The results not only replicate the findings of Swami et al. [21] but also further reveal that adult females in Malaysia have a similar level of accepting their bodies as their male counterparts do. 

Our findings benefit the literature by providing another piece of evidence to the usefulness of the English version of the BAS-2. The replication also sheds light on the qualities of the BAS-2 in the Southeast Asian context. Similarly, the findings have practical implications for body image research in Malaysia. As English is widely used across different age groups (e.g., young adults, working adults) and contexts (e.g., education, business) in Malaysia, the English BAS-2 can be used for different Malaysian populations. In the same vein, the BAS-2 can be employed in cross-cultural studies on body image. For instance, researchers may first investigate the measurement invariance of the BAS-2 across countries (e.g., [34,35]) and then employ the scale to determine whether the body appreciation level varies from one cultural sample to another, aside from examining the antecedent and consequences of body appreciation across different cultures.

### Limitations and Future Directions

The present study replicates the findings of Swami et al. [21], implying that both the English and the Malay version of the BAS-2 are useful in the Malaysian environment. However, it is premature to conclude that both versions are equivalent. Researchers who wish to use the two languages and compare the two sets of responses are reminded to examine the measurement invariance (between the two language versions). Besides, the present study omitted the test–retest reliability, construct validity, and predictive validity. Future researchers are therefore recommended to conduct a longitudinal study to further examine the test–retest reliability and predictive validity of the BAS-2. For example, researchers may use a longitudinal design to examine the impact of body appreciation on depression using the BAS-2 and the five-item World Health Organization Well-being Index (WHO-5), which has been validated for screening depression among young adults in Malaysia [36].

## 5. Conclusions

The present study demonstrates that the English BAS-2 is a psychometrically valid tool for assessing the level to which individuals in Malaysia appreciate their bodies. Both the EFA and the CFA support that the BAS-2 is a unidimensional measurement with good internal consistency, incremental validity, and measurement equivalence between genders. As obesity has become a critical issue in Malaysia, the BAS-2 would be useful for researchers and practitioners to advance the research and interventions related to body image in the Malaysian context. There is also the potential for the use of the BAS-2 in cross-cultural studies to provide valuable insights into the antecedents and consequences of body appreciation in different cultural settings.

## Figures and Tables

**Table 1 ijerph-18-11628-t001:** Summary of Exploratory Factor Analysis Results in Study 1.

Item		Factor Loading
1	I respect my body.	0.69
2	I feel good about my body.	0.82
3	I feel that my body has at least some good qualities.	0.67
4	I take a positive attitude towards my body.	0.81
5	I am attentive to my body’s needs.	0.66
6	I feel love for my body.	0.84
7	I appreciate the different and unique characteristics of my body.	0.78
8	My behavior reveals my positive attitude toward my body; for example, I hold my head high and smile.	0.69
9	I am comfortable in my body.	0.76
10	I feel like I am beautiful even if I am different from media images of attractive people (e.g., models, actresses/actors).	0.72
	Kaiser–Meyer–Olkin	0.94
	Bartlett’s test	χ^2^ (45) = 2336.69, *p* < 0.001
	Explained total variance (%)	55.90

Note. N = 392.

**Table 2 ijerph-18-11628-t002:** Goodness-of-fit Indices for Tests of Invariance of Body Appreciation Scale-2.

	*χ^2^*	df	CFI	Model Comparison	∆*χ^2^*	df	*p*	∆CFI
Baseline model								
Male	47.30	35	0.998	-	-	-	-	-
Female	29.67	35	1.000	-	-	-	-	-
1. Configural invariance model	76.97	70	1.000	-	-	-	-	-
2. Metric invariance model	133.58 ***	79	0.996	2 vs. 1	56.61	9	<0.001	0.004
3. Scalar invariance model	142.64 ***	88	0.996	3 vs. 2	9.06	9	0.432	0.000

Note. N = 797. ULS = estimator. CFI = comparative fit index; ∆CFI = difference in CFI. *** *p* < 0.001.

## Data Availability

The datasets generated during and/or analyzed during the current study are available from the corresponding author upon request.

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
