# Peer review of "Validation and Measurement Invariance of the Body Appreciation Scale-2 between Genders in a Malaysian Sample"

_ijerph, 2021, doi:10.3390/ijerph182111628_

Round 1

Reviewer 1 Report

General comment

The present study aimed to test the psychometric characteristics of the English version of the 10-item Body Appreciation Scale-2 (BAS-2) in the multilingual Malaysian context. 797 participants aged 18 to 56 years old in Malaysia that differs in races were divided into two groups to administer the English version BAS-2 and selfie editing frequency scale. The dataset submitted to exploratory factor analysis and confirmatory factor analysis respectively, the results found that replicate past findings and regularly support the usability of the BAS-2 in the Malaysian context.

The study lays a foundation for the follow-up obesity research as well as other researches related to body appreciation in Malaysia. However, the manuscript currently presents some weak points that should be addressed.

Specific comments

1 In the Introduction, the second reason to address this gap may need further explanations, for the logical relationship between sentences isn’t clear enough.

2 In the Participants and Design part, since the participants partly used snowball sampling to collect, whether the samples were sufficiently different and representative needs to be answered.

3 The use of selfie-editing frequency score is recommended to be explained in the research method, in order to clarify the research process and related purposes.

4 On line 113, it will be easier to understand if the authors add an explanation for the meaning of the names of the two sets.

Reviewer 2 Report

This manuscript examines the Validation and Measurement Invariance of the Body Appreciation Scale-2 between Genders in a Malaysian Sample.

Introduction:

  • I think it’s crucial to be clear in the Introduction about prior findings with respect to age and gender differences since a key contribution of the present manuscript seems to be based on evaluating this measure in male and female young adults and adults.

Results:

-EFA - Since BAS is an unidimentional scale, I was wondering why the authors state the use of varimax rotation method. 

-Please formate the number of decimal places according to APA style 

-CFA - include modification indexes and correlation errors

-Table 2 should include delta for Δχ2, RMSEA, and SRMR too. Since differences in the chi-square values are sensitive to small dissimilarities in covariance matrices between the groups and are also influenced by the sample size, it is suggested that, when using the Δχ2 test, results should be interpreted along with other indicators of invariance, such as ΔCFI, ΔRMSEA, and ΔSRMR. 

Round 2

Reviewer 1 Report

I would recommend publishing this manuscript.